# Derivation of Cyclic Stiffness and Strength Degradation Curves of Sands through Discrete Element Modelling

**Fedor Maksimov and Alessandro Tombari \***

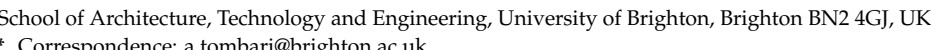

School of Architecture, Technology and Engineering, University of Brighton, Brighton BN2 4GJ, UK
\* Correspondence: a.tombari@brighton.ac.uk

**Abstract:** Cyclic degradation in fully saturated sands is a liquefaction phenomenon characterized by the progressive variation of the soil strength and stiffness that occurs when the soil is subjected to cyclic loading in undrained conditions. An evaluation of the relationships between the degradation of the soil properties and the number of loading cycles is essential for deriving advanced cyclic constitutive soil models. Generally, the calibration of cyclic damage models can be performed through controlled laboratory tests, such as cyclic triaxial testing. However, the undrained response of soils is dependent on several factors, such as the fabric, sample preparation, initial density, initial stress state, and stress path during loading; hence, a large number of tests would be required. On the other hand, the Discrete Element Method offers an interesting approach to simulating the complex behavior of an assembly of particles, which can be used to perform simulations of geotechnical laboratory testing. In this paper, numerical triaxial analyses of sands with different consistencies, loose and medium-dense states, were performed. First, static triaxial testing was performed to characterize the sand properties and validate the results with the literature data. Then, cyclic undrained triaxial testing was performed to investigate the impact of the number of cycles on the cyclic degradation of the soil stiffness and strength. Laws that can be used in damage soil models were derived.

**Keywords:** DEM; liquefaction; cyclic degradation; damage models; one-way input



## 1. Introduction

Saturated cohesionless soils, when subjected to rapid cyclic loading in undrained conditions, manifest a progressive variation of their stiffness and strength. This phenomenon is caused by the reorganization of the complex assembly of grains which tend to compress or dilate according to their initial state of compaction. If the tendency is to compress, the pore water, that cannot flow out, will contrast this change by increasing its pressure. Consequently, the contact forces between the grains will be reduced, causing soil degradation. If the excess pore pressure is high enough to reduce each contact between the particles completely, soil liquefaction will occur. Even without reaching full liquefaction, the progressive degradation of the soil properties, during the cyclic loading, strongly affects the dynamic response of their structures [1]. Therefore, an assessment of the strength and stiffness degradation of sands is essential for the calibration of cyclic damage models [2]. The simulation of this fundamental phenomenon has been extensively studied over the last decades, and a great variety of continuum constitutive models describing nonlinear aspects of soils has been proposed [3]. Because of the continuum constitutive nature of these models, an empirical or phenomenological calibration of the model parameters is required: typically, experimental triaxial tests are performed [4–6].

On the other hand, the Discrete Element Method (DEM) [7] was originally developed for soil mechanics to simulate granular assemblies with individual grains modeled by disks or spherical particles [8]. Therefore, the advantage of the DEM is the ability to simulate the material microstructure and to represent directly and intrinsically the heterogeneous discrete nature of granular soils (e.g., particle shape and size distribution) [9].

DEM simulations have been conducted to replicate geotechnical testing such as the conventional monotonic triaxial test to derive the stress–strain behavior of the numerical specimen [10] and to show typical theoretical aspects such as dilatation and critical state behavior [11–22]. Most studies are based on the use of simplified sphere particle shapes to keep calculation costs as low as possible [11,14,17–22]. On the other hand, spherical shapes can manifest excessive rolling, leading to an underestimation of the value of the friction angle as compared to real cohesionless soil [20–22]. To improve the reliability of the DEM model, an additional contribution, i.e., the rolling stiffness with an elastic contact law, has been introduced by Iwashita and Oda [21,22]; the resultant contact rolling moment acting against the relative rolling rotation of the particles increases the soil strength. This rolling resistance model has been used in [10] to simulate triaxial testing and the results evidenced the ability of the numerical model to reproduce complex behaviors, as observed in real granular materials, as the non-associative flow rule [23]. Moreover, rolling resistance can be an alternative solution to consider the particles' shapes instead of using computationally expensive polygons and polyhedrons [20,24]. Several studies have been based on the investigation of the irregularity of the sand particles' shape on the macroscopic behavior of the granular material [8–10,13,24]. In [24], a coefficient of the rolling friction related to the normalized average eccentricity of contact has been introduced to capture the effects of the grains' shape. Aboul Hosn et al. [20] performed an extensive investigation of the influence of the contact rolling resistance properties on the macroscopic behavior by performing numerical triaxial tests of sands. They showed that increasing the rolling elastic stiffness will cause an increase in the macroscopic internal friction angle; if the rolling stiffness coefficient is high enough, its influence on both the peak friction angle and the dilatancy angle becomes negligible.

The most widely used laboratory procedure to evaluate the liquefaction characteristics of sands is the cyclic triaxial test on laboratory-prepared samples [25]. The DEM has also been used to simulate fully saturated soil in undrained conditions by adopting the constant volume method [26–31]. The representation of undrained conditions through keeping volume change conditions has the advantage of saving computation time by eliminating complex fluid–particle coupling calculations. However, almost all DEM simulations of liquefiable soils focused on the micromechanical investigation of liquefaction triggering [27–31]. Martin et al. [28] focused on the micromechanical investigation of liquefaction performed by DEM simulations of cyclic undrained triaxial tests. Vinod et al. [31] claimed that a unique relationship exists between shear strain and the number of cycles for triggering the initial liquefaction.

Although the potentiality of the DEM simulation in capturing the behavior of real soils has been largely verified, the results have been rarely used to calibrate soil constitutive models to be used in practical applications, usually conducted through the Finite Element Method. Therefore, this paper aims to propose, for the first time, a derivation of the strength and stiffness degradation relationships to be used in cyclic damage models at a meso- or macro-scale, by exploiting the DEM instead of carrying out conventional but cumbersome and costly experimental testing. Numerical cyclic triaxial testing is, hence, performed, and the maximum deviatoric stress, as well as the shear modulus, is monitored with the increase of the number of loading cycles to derive these fundamental and practical relationships.

## 2. Discrete Element Method for Cyclic Triaxial Testing

The Discrete (or Distinct) Element Method pioneered by Cundall [7] is based on the numerical modelling of the contact interactions of an assembly of discrete particles that move under specifically prescribed boundary conditions. Essentially, the method entails computing the interaction forces generated by an artificial interpenetration of the particles and, subsequently, using these forces to calculate the new position of each particle through Newton's second law. A summary of the method is presented below.

### 2.1. Equation of Motion

Let us consider an assembly of $n$ particles, hereinafter embodied by circular or spherical elements, as in Figure 1. The translational motion of the $i$-th discrete element is governed by the Newton–Euler rigid body dynamics equations as follows:

$$m_i \cdot \ddot{x}_i(t) = F_i(t), \tag{1}$$

where $\ddot{x}_i$ is the acceleration of the center of mass of the particle, $m_i$ is its mass, and $F_i$ is the resultant force vector expressed as follows:

$$F_i(t) = \sum_{c \in C^i} F_i^c(t) + F_i^{ext}(t) + F_i^{damp}(t), \tag{2}$$

In Equation (2), $F_i^c(t)$ is the vector of the contact forces, acting at the set of the $m$ contact points, $C^i = \{c_1, c_2 \ldots c_m\}$, with each one corresponding to the middle of the artificial overall zone between any two particles (see Figure 1); $F_i^{ext}$ is the external load vector, acting on the $i$-th particle at the time $t$, and $F_i^{damp}$ is the global damping force vector. It is worth mentioning that $F_i^{ext}(t)$ represents the body forces, such as the gravitational force component $f_i^{ext} = m_i \cdot g$, where $g$ is the gravity acceleration; this contribution can be neglected for quasi-static triaxial analysis, and they will not be considered in this study.

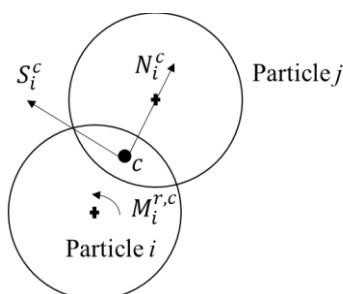

**Figure 1.** Contact forces generated acting on the $i$th particle at contact point $c$.

The contact force vector $F_i^c(t)$ at the contact point, $c$ in Equation (2), can be decomposed into the normal and tangential or shear components, $N_i^c$ and $S_i^c$, respectively:

$$F_i^c(t) = N_i^c + S_i^c \tag{3}$$

The tangential or shear force, $S_i^c$, contributes to the sliding motion between two interacting particles, as well as to the rotation about its center of mass; this rotation allows the rolling motion of the particles relative to each other, which is a fundamental aspect for the correct simulation of the redistribution of the grains in a soil sample [12,19,20]. Therefore, the rotational motion of a spherical particle of radius $r_i$ can be described by the following expression:

$$I_i \cdot \dot{\omega}_i(t) = M_i(t), \tag{4}$$

in which $\dot{\omega}_i$ is the angular velocity of the center of mass of the particle, $I_i$ is the moment of inertia tensor of components $I_i = {}^2\!/_5 \cdot m_i \cdot r_i^2$ about its center of mass, and $M_i$ is the resultant moment. The resultant moment, $M_i(t)$ in Equation (4), is derived from the cross product ($\times$) of the tangential contact forces, $F_t$, by the unit normal vector, $n_c$, perpendicular to the contact's tangent plane, as follows:

$$M_i(t) = \sum_{c \in C^i} M_i^{r,c}(t) + \sum_{c \in C^i} s^c \times F_i^c(t) + M_i^{damp}(t), \tag{5}$$

where $s^c$ is the vector connecting the center of the $i$-th particle to the contact point $c \in C^i$, $M_i^{r,c}(t)$ is the rolling moment vector at the same contact points, and $M_i^{damp}$ is the global damping moment.

Equations (1) and (4) can be solved through an explicit time integration scheme, such as the "leap-frog" algorithm [32], obtained by modifying the Verlet difference scheme [33]. Therefore, at each time, *t*, new particle positions are computed, and a new contact or collisions are detected or updated; this generates new interaction forces that will be used in Equations (1)–(5) to reach a new state of equilibrium.

In this paper, the simulation of the cyclic triaxial testing was performed under quasi-static conditions; therefore, to facilitate a rapid convergence, a purely numerical global non-viscous damping was used [34],

$$F_{damp} = -\alpha_{damp} \cdot sgn(v_i)|F_i| \tag{6}$$

and

$$M_{damp} = -\alpha_{damp} \cdot sgn(\dot{\omega}_i)|M_i| \tag{7}$$

where $\alpha_{damp}$ is the positive numerical damping coefficient, and $sgn(\bullet)$ returns the sign of the *i*-th component of the translational velocity, $v_i$, and angular velocity, $\dot{\omega}_i$.

## 2.2. Contact Model

The Cundall-Strack elastic perfectly brittle contact model [34] was used to determine the normal and tangential contact forces in Equation (3). Therefore, the normal force acting at the *i*-th particle induced by the artificial overlap with the *j*-th particle, $d_{ij}$, is given by the following expression:

$$N_i^c = k^n d_{ij} n_c^i \tag{8}$$

where $k^n$ is the normal stiffness and $n_c^i$ is a unit normal vector that is perpendicular to the contact's tangent plane. The shear force vector, obtained by summing up all the incremental tangential contributions, is given by the following expression:

$$\Delta S_i^c = -k^s \Delta U_{ij}, \tag{9}$$

where $k^s = v k^n$ is the shear stiffness related to the normal stiffness through the Poisson's ratio, $v$, and $\Delta U_{ij}$ is the relative tangential displacement at the contact point. The incremental formulation is required to implement the Mohr–Coulomb rupture criterion, which characterizes the typical behaviour of non-cohesive geomaterials:

$$\|S_i^c\| \leq \|N_i^c\| \tan\mu, \tag{10}$$

where $\mu$ is the interparticle angle of friction, and $\|\bullet\|$ is the norm operator. The rolling moment, $M_i^{r,c}$, in Equation (5) represents the rolling resistance against the relative rolling rotation, $\theta_{ij}$, of two particles; it can be computed as the sum of the incremental contributions expressed as follows:

$$\Delta M_i^{r,c} = -k^r \Delta\theta_{ij}, \tag{11}$$

in which $k^r$ is the rolling stiffness defined as follows:

$$k^r = \alpha \cdot k^s \cdot r_i \cdot r_j \tag{12}$$

In Equation (12), $r_i$ is the radius of the *i*-th particle, and $\alpha$ is the rolling stiffness coefficient. As for the shear force, the rolling resistance can be expressed through the frictional law:

$$\|M_i^{r,c}\| \leq \|N_i^c\| \eta^r \min(r_i, r_j), \tag{13}$$

where $\eta^r$ is the limiting rolling coefficient.

## 2.3. Calibration of the Input Parameters for Cyclic Triaxial Testing

The contact model described in Section 2.2 can be fully determined by six parameters: the mass of each particle, $m_i$, the normal stiffness, $k^n$, the Poisson's ratio, $v$, the interparticle

angle of friction, $\mu$, the rolling stiffness coefficient, $\alpha$, and the limiting rolling coefficient, $\eta^r$. It should be noted that these parameters (except for the particle mass) do not have a direct physical relation with the microscopic properties of a real soil grain. Nevertheless, as observed in several past studies [12–15,20], a few of these parameters can be considered to have a secondary impact on the macroscopic behavior of an assembly of particles [12,20] if a few dimensionless indices are opportunely calibrated; for instance, it can be defined a dimensionless stiffness level [20]:

$$k = \frac{k^n}{2r \cdot p}, \tag{14}$$

where $p$ is the confining pressure, which in our case, should be set greater than 1000 to have a negligible effect on the elastic properties of the macroscopic behavior [35]. The particle mass can be adjusted to verify two conditions; the first is related to the inertial number as follows:

$$I = \dot{\varepsilon} \cdot 2r \sqrt{\frac{m_i}{V_s p}}, \tag{15}$$

in which $\dot{\varepsilon}$ is the strain rate of the boundary conditions used to simulate the cyclic triaxial testing, and $V_s$ is the volume of the sphere. In an ideal quasi-static condition, $I = 0$; Radjai and Dubois [36] suggested that the quasi-static limit is approached for $I < 10^{-4}$. The second condition is aimed to reduce the computational cost through the upper limit of the time step in order to ensure the stability of the explicit integration scheme through the following relation:

$$\Delta t_{min} = r \cdot \sqrt{\frac{2r\rho}{k^n}}, \tag{16}$$

where $\rho$ is the particle density linked to the mass, $m_i$. Therefore, the mass of each particle could be set as high as possible in order to reduce the computational cost of the simulation if the quasi-static condition is met by limiting the inertia number of Equation (15). Thornton and Antony [37] suggested increasing the particle density up to a fictitious $10^{12}$ kg/m³ value, whilst Macaro and Utili [38] adopted a value of $10^9$ kg/m³ for the simulation of triaxial tests of seabed sands. On the other hand, the primary parameters are related to the assembly's frictional behavior, such as the interparticle angle of the friction, $\mu$, and the limiting rolling coefficient, $\eta^r$. In particular, the latter has the strongest influence on the peak strength, the residual or critical strength, and the contractive/dilative behavior [20]. These two parameters are calibrated in this paper to define a macro-category of soil whose physical characteristics are consistent with the average properties obtained from real sand with uniform grains. Therefore, the Discrete Element Method is able to capture several complex constitutive behaviors with few parameters; the final behavior will be essentially affected by the initial void ratio and the distribution of the grains of the soil sample, as shown in the following Section 3.

## 3. Results

In this section, the results of the cyclic triaxial testing are presented in order to propose a methodology to derive the numerical stiffness and strength degradation relation that can then be used at the mesoscale level to calibrate constitutive models or at a macroscale to calibrate macro-models such as those formulated through the dynamic p-y curve approach [1]. First, monotonic drained triaxial testing was conducted in order to obtain the geotechnical properties of the two investigated samples, and, then, undrained cyclic triaxial testing was performed to derive the degradation curves. The simulation was conducted by the open-source code, YADE [39].

### 3.1. Model Description

In this application, two soil samples featuring the behavior of loose sand and medium-dense sand are modelled through the Discrete Element Method, as shown in Figure 2. The loose packing of the spheres is characterized by a relative density of $D_r = 33\%$ and

a void ratio of $e = 0.72$; the medium-dense sample has a relative density of $D_r = 55\%$ and a void ratio of $e = 0.60$. The two soil samples are representative of uniformly graded, well-rounded, coarse sand. Numerical simulations have been conducted on a representative cell with periodic boundary conditions. In order to satisfy periodicity conditions, a periodic space is created by the repetition of a parallelepiped-shaped cell. The use of the periodic boundary allows for the production of homogeneous and isotropic states, eliminating the disturbances in the granular structure induced by the wall effect [12]. The three-dimensional assemblies are composed of 20,199 spheres for the loose sand sample and 20,898 for the medium-dense sample, having a mean particle radius of 1 mm (hence, representing coarse sand); the radii, ranging from a minimum of 0.92 mm to a maximum of 1.09 mm, are drawn randomly from a uniform distribution.

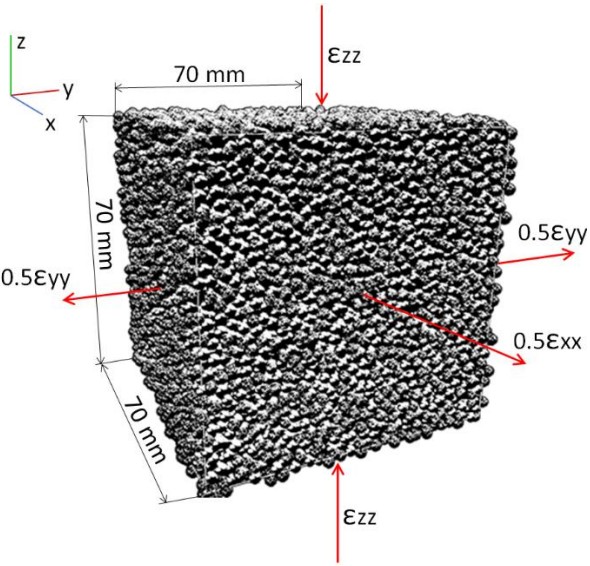

**Figure 2.** Assembly of particles used to simulate the behavior of the soil samples.

This approach has been used to avoid the phenomenon of "crystallization", which occurs with the perfectly regular packing of particles with the same radius, yielding an unrealistic, extremely stiff, and strong soil medium. The relative density, $D_r$, is obtained as follows:

$$D_r = \frac{e_{max} - e}{e_{max} - e_{min}}, \tag{17}$$

where $e$ is the current void ration, and $e_{max}$ and $e_{min}$ are the maximum and minimum void ratios, respectively. For uniform spheres with the same radius, $e_{max} = 0.90986$, and $e_{min} = 0.35047$ [40]. The current void ratio is expressed as

$$e = \frac{u_s}{1 - u_s} \tag{18}$$

in which the porosity of the sample, $u_s$, is numerically computed from the subtraction of the total volume of the cell and the total volume of the rigid spheres.

The input parameters used in this study are reported in Table 1, defined as described in Section 2.3, and selected in the range of values proposed in [20] to simulate loose and medium-dense sands. The preparation procedure consists of an isotropic compression of a randomly generated packing of floating spheres as follows: (1) initially, the spheres are generated randomly and placed within a cube of a 0.07 m side in order to avoid any contact between them; (2) the packing is subjected to an isotropic compression by moving all the sides (boundary conditions) of the cube uniformly until the target value of the relative density has been reached; (3) the overlaps are eliminated by slightly decreasing the particles' radius by a factor of 0.999 in order to obtain an initial stress-free packing.

**Table 1.** Input parameters selected for the numerical simulations.

| Parameters | Symbols | Values |
|---|---|---|
| Dimensionless stiffness level | $k$ | $5 \times 10^6$ |
| Poisson's ratio | $\nu$ | 0.2 |
| Rolling stiffness coefficient | $\alpha$ | 2 |
| Particle density | $\rho$ | $1 \times 10^{12}$ kg/m³ |
| Interparticle friction angle | $\mu$ | 30° |
| Limiting rolling coefficient | $\eta^r$ | $10^{-2}$ |
| Mean particles' radius | $r$ | $10^{-3}$ m |

*3.2. Monotonic Triaxial Testing*

After the preparation procedure to obtain the target initial relative density, the two specimens are subjected to drained triaxial compression to derive the main geotechnical properties, such as the angle of internal friction and shear modulus. A monotonic triaxial test comprises two phases: (i) isotropic consolidation and (ii) the deviatoric stage. A constant loading strain rate of 0.0001/s is applied to the boundary conditions in order to avoid inertial effects and maintain quasi-static conditions; the inertial number calculated through Equation (15) is equal to $1.65 \times 10^{-4}$; hence, smaller than $10^{-3}$, as aimed. In this study, the timestep used for the analyses is set to 0.01481 s, corresponding to 50% of the limit value defined by Equation (16).

During the isotropic stage, the strain-control conditions are applied (by uniformly moving the artificial sides of the cube) to achieve constant isotropic stress equal to 100 kPa. Subsequently, a deviatoric phase is carried out by moving the upper and bottom periodic boundaries at the constant loading strain rate while controlling the lateral sides in order to maintain the confining stress of 100 kPa. The frictional angle is calculated through the following expression:

$$\varphi = \sin^{-1}\left(\tan\left(\frac{q}{2p}\right)\right), \tag{19}$$

where q and p are the deviatoric and the mean stress computed at the peak or at the critical state, respectively.

Calculated geotechnical parameters have been summarized in Table 2. The results on the strength and stiffness evidence a good agreement with the experimental data for loose and medium coarse-grained sand [41,42]; therefore, the discrete element model is able to capture the global behavior of the soil at the mesoscale. Figure 3 illustrates the stress–strain and volume change behaviors of the two samples in loose and medium-dense states. A maximum value of an axial strain of 20% is applied to reach a critical state. Figure 3a shows that at large strains both loose and medium-dense soils reach the same critical shearing resistance characterized by the residual angle of friction of 27° and the same void ratio, termed the critical void ratio ($e = 0.75$), corresponding to a critical relative density of 29%. This is consistent with the critical state concept [23] in which cohesionless soil tends to move toward the critical void ratio regardless of its initial value of the void ratio. The critical void ratio marks the boundary between a contractive and a dilative response; therefore, it is expected that a very small dilatancy occurs for the loose sand sample ($e = 0.72$) and a much larger dilation for the medium-dense sample ($e = 0.60$). Moreover, it can be noted that the sample at medium-dense density mobilizes a peak shearing resistance of 35°, which is much higher than the residual fraction angle of 27°, whilst an almost constant deviatoric plateau is obtained for the loose soil sample. The evolution of the volumetric strain with the axial strain is shown in Figure 3b; the increment of the volumetric strain indicates an increment of the total volume of the specimen and, hence, a dilative behavior, which is higher in medium-dense sands than in loose sands. For small values of axial strain, a contractive behavior is observed.

**Table 2.** Results of the monotonic triaxial testing.

| Parameters | Loose Sand | Medium-Dense Sand |
|---|---|---|
| Relative density, $D_r$ [%] | 33 | 55 |
| Rel. density at critical state [%] | 29 | 29 |
| Peak friction angle [°] | 30 | 35 |
| Residual frictional angle [°] | 27 | 27 |
| Initial Shear Modulus [MPa] | 34 | 49.5 |

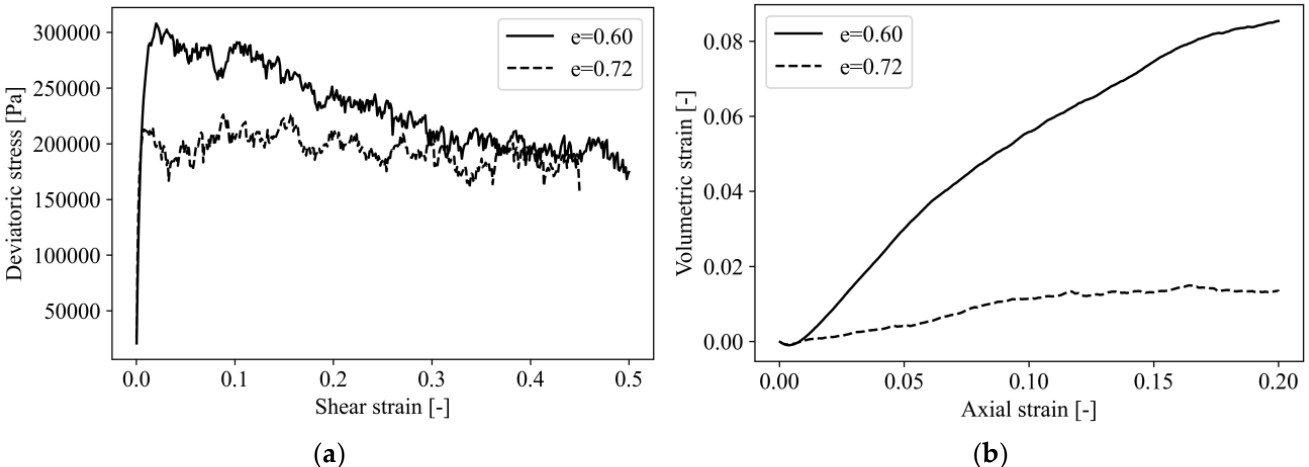

(**a**)                    (**b**)

**Figure 3.** Results of the numerical drained triaxial testing for the loose and medium-dense soil specimens: (**a**) stress–strain curve; (**b**) volumetric strain behavior.

Figure 4 shows the degradation curve of $G/G_{max}$ versus the shear strain. The tangential shear modulus, $G_{max}$, is defined at a strain level of $10^{-4}$. Typically, the normalized shear modulus, $G/G_{max}$, decreases with the increase of the shear strain, and it can be observed that the loose sand shows a stronger nonlinearity than the medium-dense sand, as expected. These curves can be used for typical problems of seismic site response analysis where the definition of an equivalent linear model for the soil is required [43].

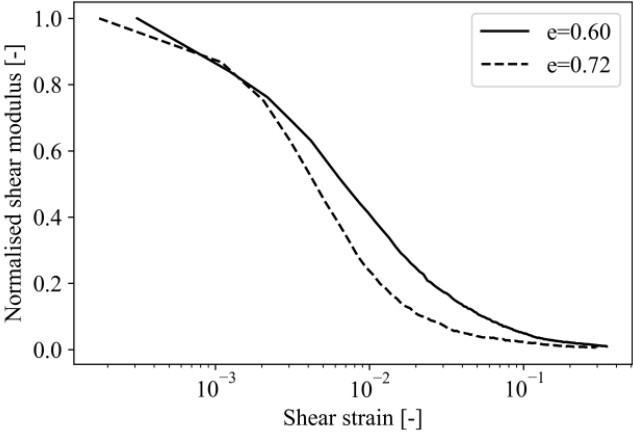

**Figure 4.** Shear modulus degradation during monotonic drained triaxial tests for loose and medium-dense packing.

### 3.3. Cyclic Triaxial Testing

The undrained cyclic triaxial test is carried out by applying a one-way strain-control input, as shown in Figure 5, for the loose and medium-dense sand, respectively; different maximum axial strain amplitudes, $\varepsilon_{zz}^{max}$, ranging from 0.1% to 0.5% for loose soil and from

0.1% to 5% for medium-dense soil, are applied to identify the effect of the magnitude of the input (related to the geotechnical concept of the cyclic stress ratio) on the degradation as a function of the number of cycles. The same soil specimens of the input parameters in Table 1, tested under monotonic conditions, are now considered for cyclic testing. The first stage of isotropic consolidation is performed under the same drained conditions of the previous test, reaching a confining pressure of 100 kPa. On the other hand, to perform the undrained shearing stage, a constant volume approach is used [26]; instead of simulating the interaction of the solid particles with the fluid occupying the pores of the sample, this approach aims to simulate the main kinematic condition arising from an undrained test, i.e., the shearing deformation occurs, preserving the total volume.

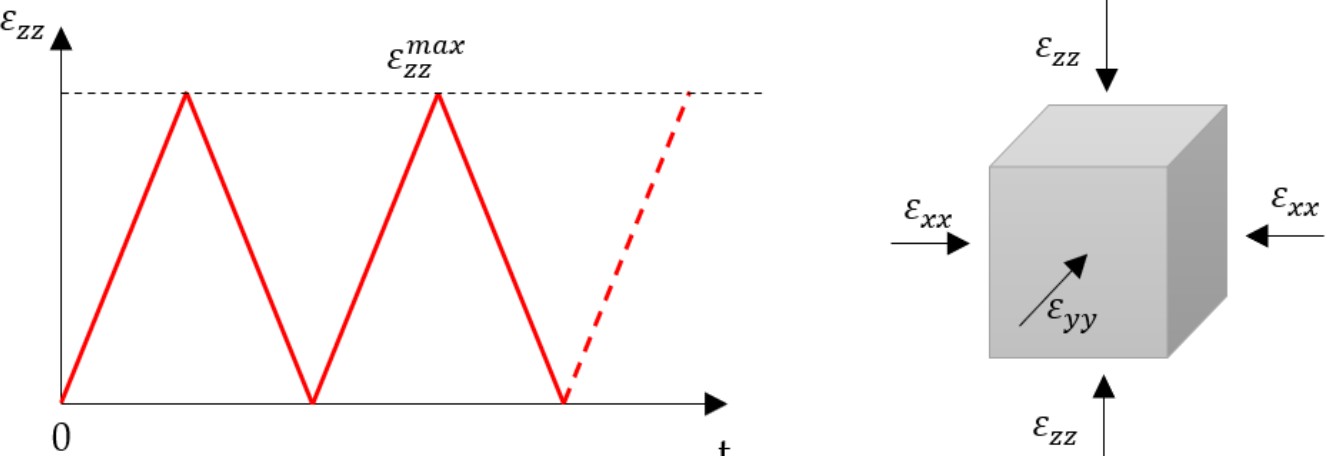

**Figure 5.** One-way strain-controlled input for cyclic triaxial testing.

Therefore, this condition can be achieved by controlling the prescribed strain of the boundaries in order to maintain the following relation:

$$\varepsilon_{xx} = \varepsilon_{yy} = -0.5 \cdot \varepsilon_{zz}, \tag{20}$$

in which $\varepsilon_{xx}$ and $\varepsilon_{yy}$ are the strains of the lateral boundaries of the cell, whilst $\varepsilon_{zz}$ is the axial strain of the top and bottom boundary controlled during the triaxial testing, according to the one-way input of Figure 5.

This method reduces the computational complexity of a solid–fluid couple model, and it is appropriate for simulating the undrained behaviors of granular materials with coarse grains [26], as were the soil samples investigated in this paper.

Figure 6a shows the hysteresis curves obtained from the results of the cyclic triaxial testing for the loose sand at several levels of maximum strain. The hysteresis curves show a typical oval-shaped behavior characteristic of loose sand [44]; the area covered by each loop, related to the dissipation of energy, increases with the increase of the level of strain. It can be observed that the hysteresis loops rotate after each cycle, manifesting an important degradation of the soil strength; in real loose sand, this phenomenon is induced by the increment of the excess pore pressure caused by the compaction of the soil grains, which decreases the effective soil stresses. This effect has been properly captured, as shown in Figure 6a, by the numerical simulation conducted under the constant volume condition. To assess the strength degradation, Figure 6b depicts the evolution of the deviatoric stress with the progression of the cyclic test, i.e., the number of cycles. The maximum value of deviatoric stress decreases after each cycle: for a small level of strain above 0.2%, a few cycles are needed to reach a level of liquefaction where the soil loses its strength entirely.

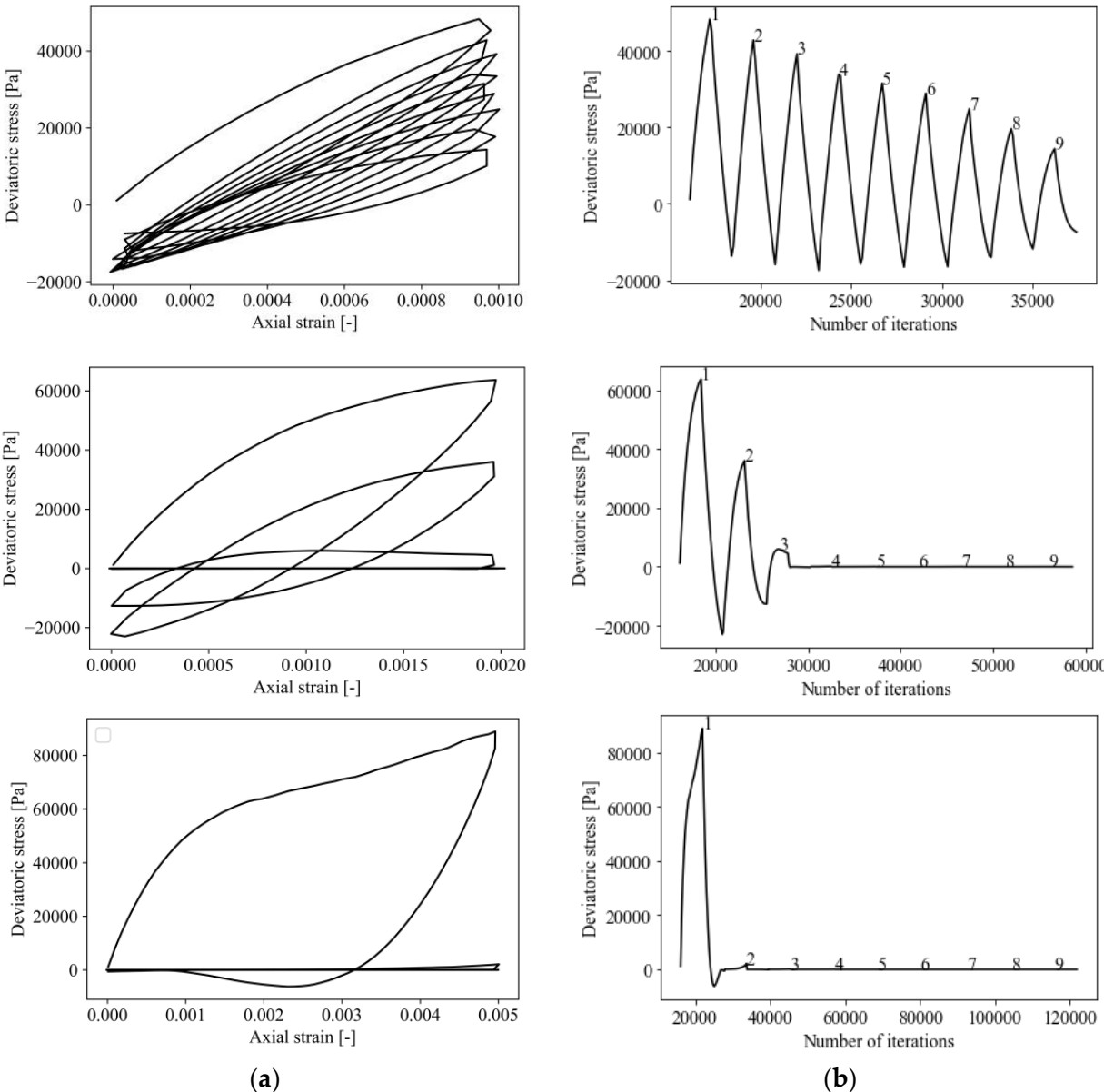

**Figure 6.** Results of the cyclic triaxial testing for loose sand: (**a**) deviatoric stress–strain hysteresis curves; (**b**) evolution of the deviatoric stress with the increase of the number of cycles.

The variation of the deviatoric stress with the number of cycles is due to the re-organization of the granular material and, hence, to the different exchange of contact forces between the particles; the evolution of the normal contact forces for a representative volume element of the loose sand sample is shown in Figure 7. A little variation of the maximum magnitude of the contact forces is observed at a small level of axial strain during the first three cycles; on the other hand, with the increase of the maximum level of strain, the degradation of the maximum value of the contact forces becomes relevant after a few cycles. It can also be observed that the number of interactions between the particles decreases with the increase of the number of cycles, similarly to what occurs in real soil during a liquefaction phenomenon.

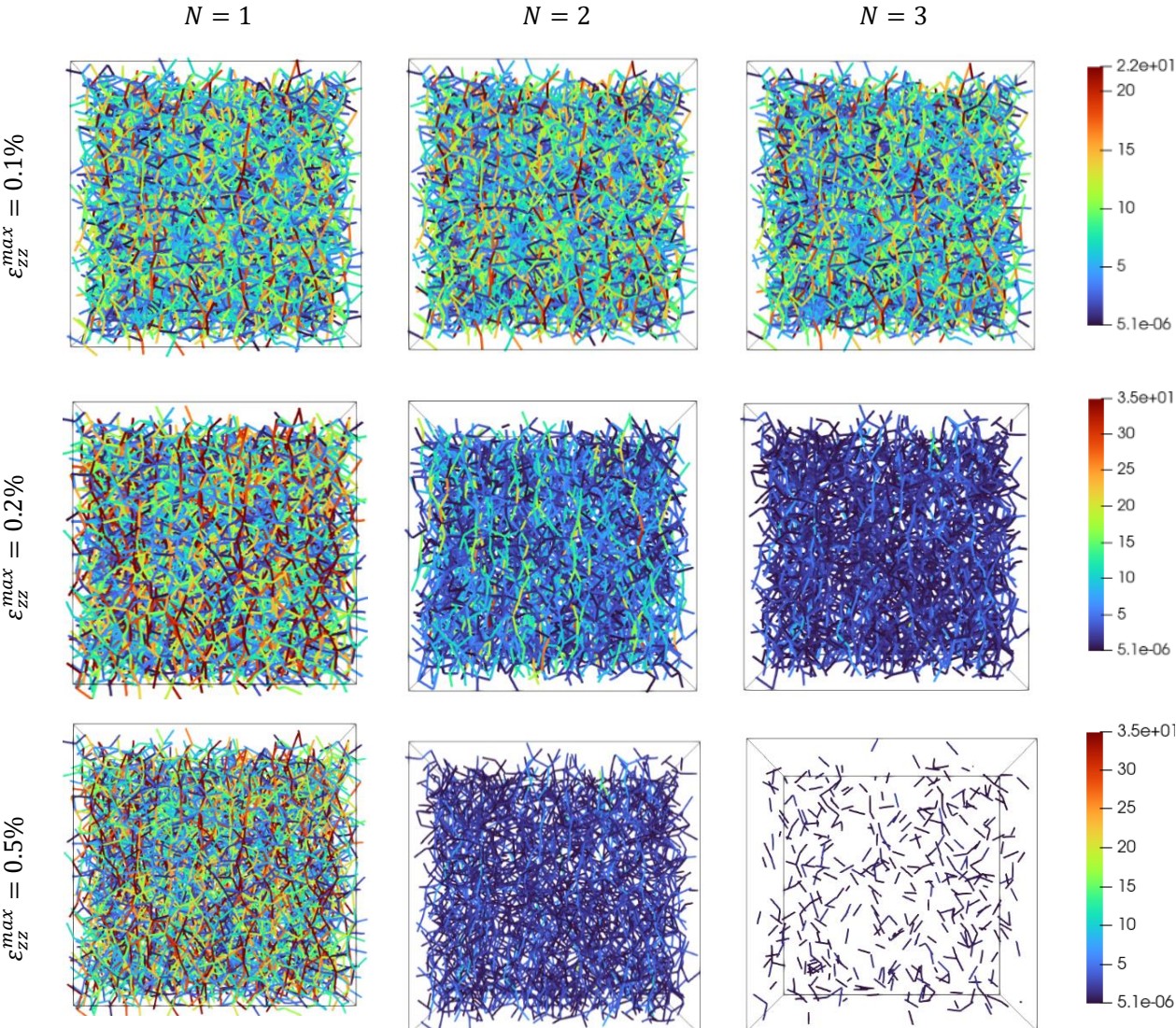

**Figure 7.** Normal contact forces for a representative volume element obtained from the results of the cyclic triaxial testing for loose sand at various maximum axial strain levels and at an increasing number of cycles (values in Newton).

Figure 8a illustrates the deviatoric stress–strain cyclic response for medium-dense soil, and Figure 8b shows the evolution of the deviatoric stress with the progress of the cyclic testing (*Dr* = 55%). It is interesting to note the different shapes of the medium-dense samples compared to the loose ones: they present the typical "S shape" or "banana shape" observed in real experiments [45,46]. The change in the stiffness and strength response with the increase of the strain level corresponds to the tendency to move from the volume contraction to the volume dilation.

Remarkably, the samples subjected to low axial strain corresponding to 0.1–0.5% show an important and gradual loss of shear resistance whilst at larger strain levels, the hysteresis loops after an initial partial degradation and become stable, showing a behavior similar to the plastic shakedown of the traditional elastoplastic theory. Moreover, the cyclic test with medium-dense samples reveals a general strain hardening type of behavior: while at low strain levels, the contraction induces a partial degradation, and at a large strain, dilatancy helps to reduce the effect of the cyclic strength degradation.

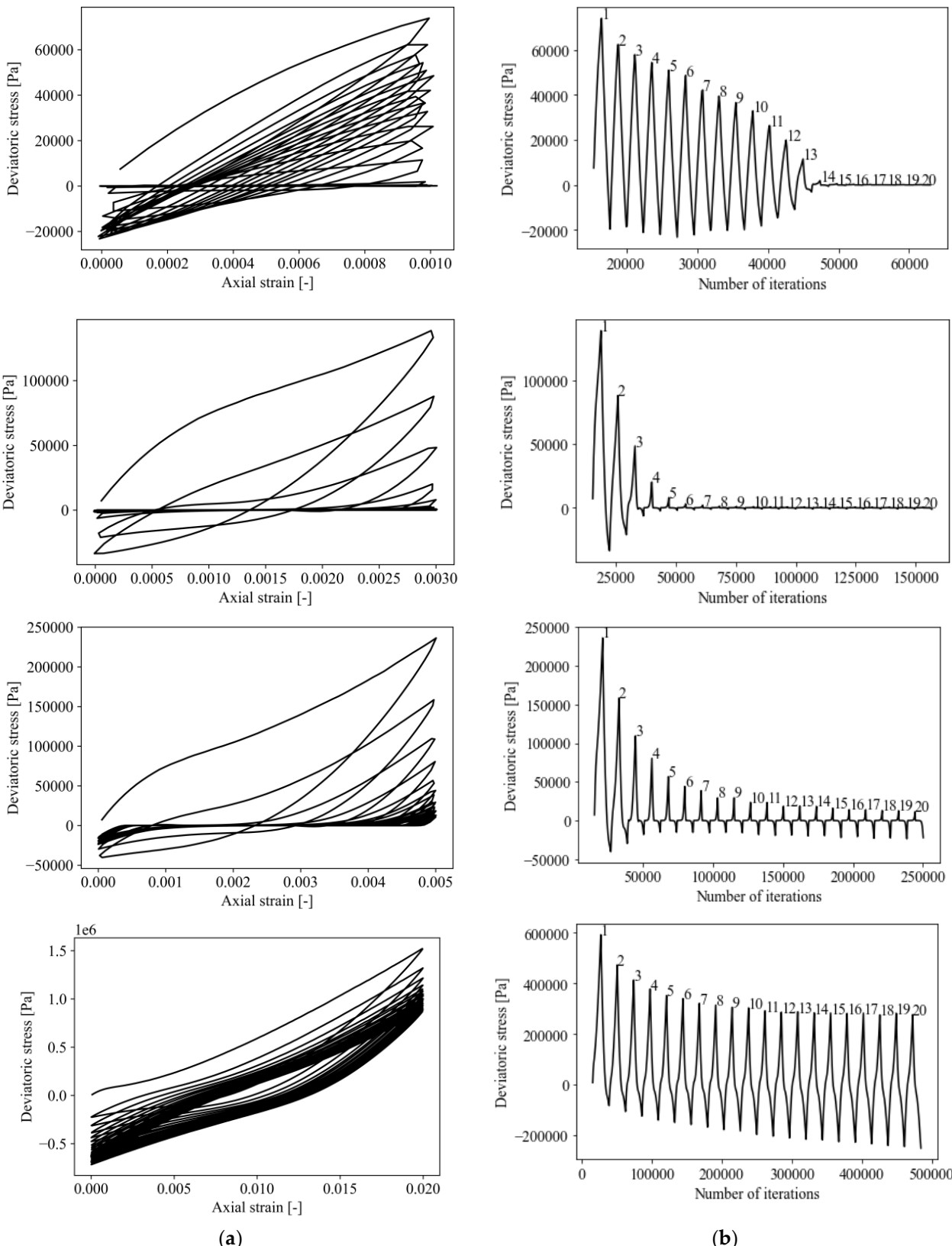

**Figure 8.** Results of the cyclic triaxial testing for medium-dense sand: (**a**) deviatoric stress–strain hysteresis curves; (**b**) evolution of the deviatoric stress with the increase of the number of cycles.

## 4. Discussion

In this section, the results of the cyclic triaxial testing are elaborated to derive the cyclic stiffness and strength degradation curves. Adopting the general fatigue-based approach, these derived curves can be used in models based on the total stress approach where the cyclic degradation can be expressed as a function of the number of loading cycles [47–49].

Figure 9a shows, for the loose sand samples, the evolution with the number of cycles of the Strength Degradation Index, $\delta_N^q$, defined as:

$$\delta_N^q = \frac{q_1^{max}}{q_N^{max}},\tag{21}$$

where $q_1^{max}$ is the maximum deviatoric stress recorded at the first loop and $q_N^{max}$ is the maximum deviatoric stress recorded after $N$ strain-controlled loops. It can be observed that the soil strength rapidly decreases with the number of cycles and with the increase of the maximum axial strain, $\varepsilon_{zz}^{max}$. The authors proposed to consider soil failure when the strength degradation index, $\delta_N^q$, is lower than 0.1. Moreover, it can be observed that the almost linear degrading behavior of the strength before the soil fails.

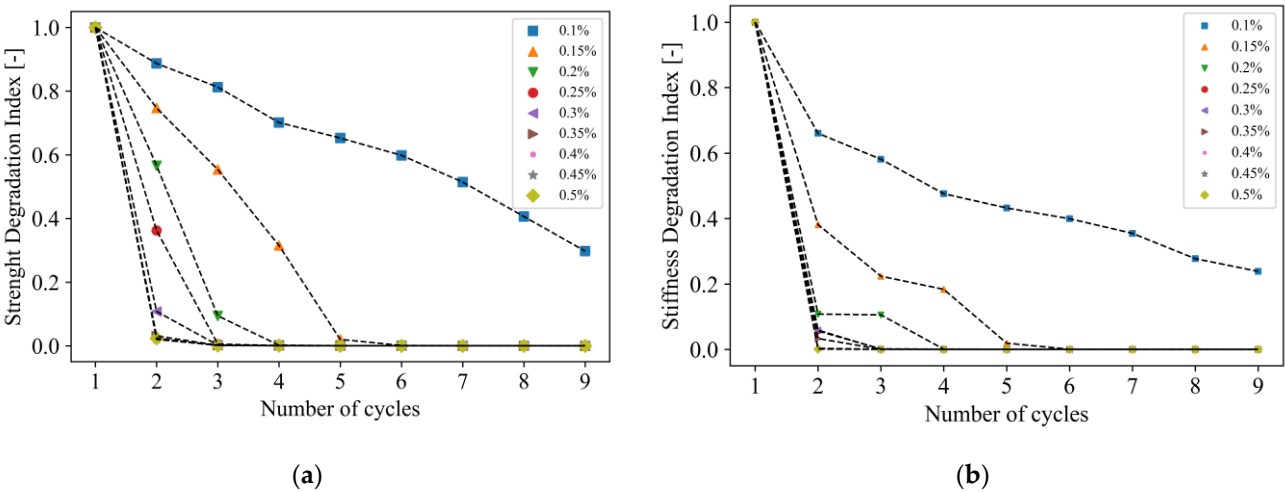

**Figure 9.** Cyclic Degradation Curves for the loose sand samples: (**a**) degradation of the soil strength; (**b**) degradation of the soil stiffness expressed as the initial tangent shear modulus.

Figure 9b shows the Stiffness Degradation Index, $\delta_N^G$, for the same results on loose sand samples, defined as

$$\delta_N^k = \frac{G_1^{init}}{G_N^{init}},\tag{22}$$

where $G_1^{init}$ is the initial tangent shear modulus calculated at the first loop, and $G_N^{init}$ is the initial tangent shear modulus at the current N loop. Similar to the strength degradation index, the curves show the progressive degradation of the stiffness with the increase of the number of cycles and strain level.

Figure 10a shows the Strength Degradation Index from the results of the medium-dense sand samples. The failure due to the soil liquefaction ($\delta_N^q < 0.1$) occurs at low strain levels when the soil manifests a compressive behavior; this leads to a fast pore pressure buildup and, consequently, a degradation of the soil strength. On the other hand, because of the relevant dilative behavior that occurs at larger strain levels, the degradation of the soil strength is limited, and the samples do not reach a failure condition. This phenomenon affects the tangent stiffness value at each loop; at low strain levels, a softening behavior is observed whilst cyclic hardening is manifested at a large strain with relevant increments, as evidenced in Figure 10b.

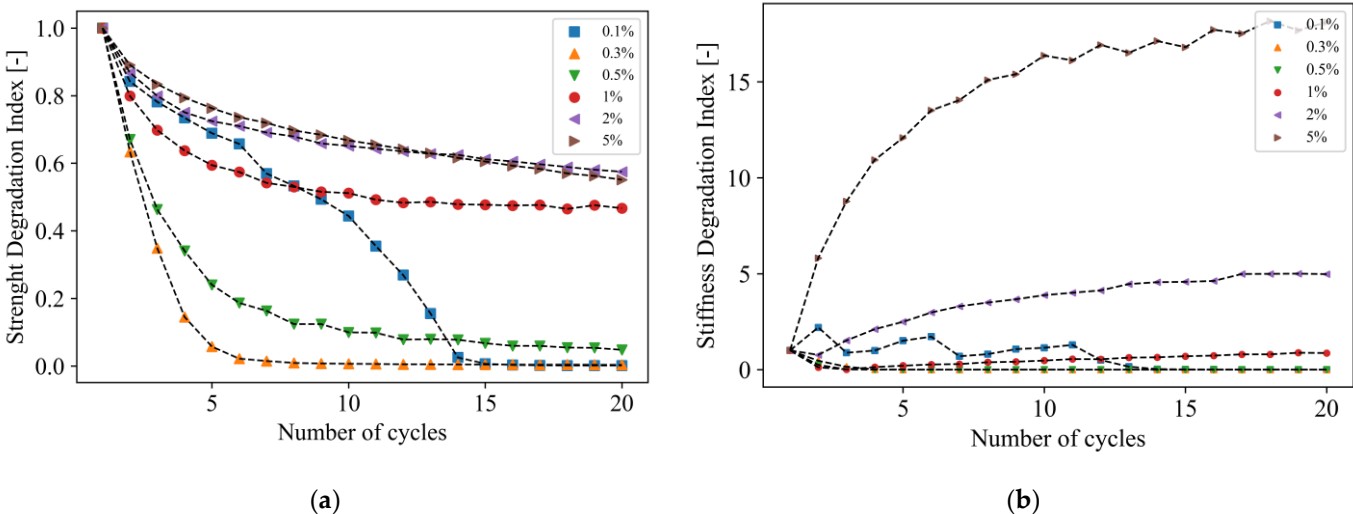

(**a**)                                                           (**b**)

**Figure 10.** Cyclic Degradation Curves for the medium-dense sand samples: (**a**) degradation of the soil strength; (**b**) degradation of the soil stiffness expressed as initial tangent shear modulus.

Finally, the results presented above are elaborated to obtain useful laws for soil damage models. A linear degradation model could be used to capture the relation between the Strength Degradation Index and the number of cycles, as depicted in Figure 11a,b, for loose and medium-dense sand, respectively. Considering the samples in which the failure occurred, the following relation is proposed:

$$\delta_N^q = 1 + m(N_c - 1), \tag{23}$$

where $N_c$ is the current number of cycles, and $m$ is an empirical coefficient depending on the maximum level of strain, $\varepsilon_{zz}^{max}$, which can be expressed through the following power law:

$$m = -\frac{0.9}{1 - a(\varepsilon_{zz}^{max})^b}, \tag{24}$$

in which $a = 2.305^{-4}$ and $b = -1.177$ for loose sand, and $a = 0.2241$ and $b = -0.59337$ for medium-dense sand.

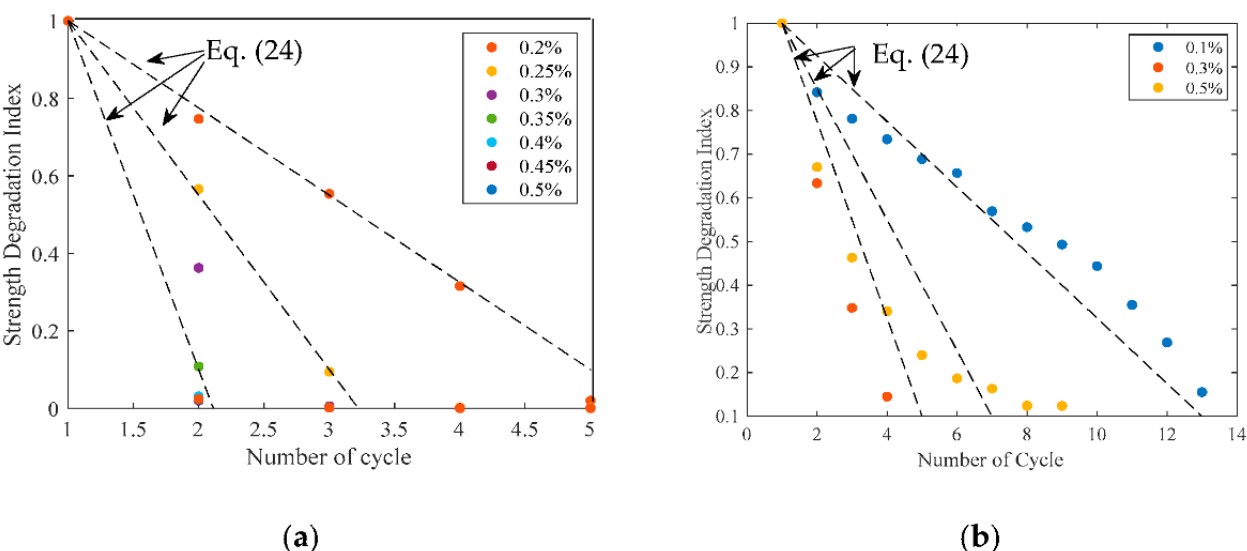

(**a**)                                                           (**b**)

**Figure 11.** Linear Strength Degradation Model for several maximum levels of strain: (**a**) loose sand; (**b**) medium-dense sand.

## 5. Concluding Remarks

The potential of the discrete element method in simulating the kinematics of an assembly of particles has been exploited in this study to investigate the cyclic behavior of two ideal soil samples representing uniform sand at two different levels of compaction, a loose and a medium-dense state. Few parameters have been used to create the model; the primary ones are the limiting rolling coefficient, which simulates the ability of a particle to roll over the other and, hence, is somehow related to the texture of the soil grains, and the void ratio of the packing; the secondary ones, such as particle density or interparticle stiffness, are fixed according to few computational rules. First, the samples have been tested to characterize the main geotechnical parameters, such as the peak and residual angle of friction, as well as the shear modulus. Second, cyclic triaxial testing was carried out by adopting a constant volume approach to replicate undrained conditions. The degradation of the deviatoric stress and shear modulus has been monitored to obtain a relation with the number of cycles applied to the sample. Briefly:

1.  Geotechnical properties such as the peak and residual angle of friction, as well as the shear modulus, are consistent with the values given in the literature from real soils.
2.  The different volumetric behavior of loose sand and medium-dense sand has been simulated without resorting to complex constitutive models or the calibration of several parameters. The same input parameters were used for both specimens, evidencing how the distribution of the particles and, hence, the void ratio of the packing is the main aspect affecting the soil behavior of the sands.
3.  The cyclic contractive tendency of the loose soil under undrained conditions at small values of strain observed during the monotonic triaxial testing leads to a cyclic degradation of its strength and stiffness; this is compatible with the real phenomenon of the liquefaction or cyclic mobility.
4.  The cyclic tendency to dilate in medium-dense sands yields to a stabilization of the soil degradation; this is compatible with the reduction of the excess pore pressure, as occurred in the real medium-dense sands.
5.  The coefficients to model a linear cyclic damage model were obtained as a function of the maximum applied axial strain and soil consistency.

Whilst the results presented in this paper are limited to a simplified model of uniform sands, the proposed investigation can be used to simulate a multitude of packing of soils at different states of compaction and dispersion of the grains. Therefore, this approach can be adopted to derive statistical relations between the cyclic soil degradation or hardening and the number of cycles, to help with the calibration of geotechnical models at the meso- or macro- scale.

**Author Contributions:** Conceptualization, A.T.; methodology, A.T. and F.M.; software, F.M.; validation, A.T. and F.M.; formal analysis, A.T.; investigation, F.M.; resources, A.T. and F.M.; data curation, F.M.; writing—original draft preparation, A.T. and F.M.; writing—review and editing, A.T. and F.M.; visualization, F.M.; supervision, A.T.; project administration, A.T.; funding acquisition, A.T. All authors have read and agreed to the published version of the manuscript.

**Funding:** This research received no external funding.

**Institutional Review Board Statement:** Not applicable.

**Informed Consent Statement:** Not applicable.

**Data Availability Statement:** Not applicable.

**Conflicts of Interest:** The authors declare no conflict of interest.

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
