# Peer review of "Derivation of Cyclic Stiffness and Strength Degradation Curves of Sands through Discrete Element Modelling"

_2673-3951, doi:10.3390/modelling3040026_

Round 1
Reviewer 1 Report
Discrete element simulation of triaxial analyses of sands with different consistencies, 17 loose and medium-dense states, have been performed in this manuscript.
It should be pointed out that in the past two decades, there have been too many DEM numerical simulation studies of granular materials, and the innovation of this paper is not enough. In addition, as a numerical simulation paper, there is only one figure about DEM model , and there is no meso analysis of particle movement and force chain development, and so on. Therefore, this article was rejected.
Reviewer 2 Report
In this paper, numerical triaxial analyses of sands under cyclic undrained triaxial testing are simulated with DEM. The main problem with this manuscript is that when it comes to cyclic loading for an undrained test, the development of pore water pressure is a critical item.
In this paper, instead of simulating the interaction of the solid particles with the fluid occupying the pores of the sample to perform the undrained shearing stage, a constant volume approach is used. I am not sure exactly whether this approach works or not. Maybe it is better to demonstrate the error of this method.
Reviewer 3 Report
This manuscript is proposing a discrete element model (DEM) to simulate the meso-scale behaviors of saturated sand under monotonic loading and cyclic triaxial loading conditions, aiming to characterize the macroscopic properties, e.g. strength and modulus degradation induced by cyclic shear loading and extracting the pattern of degradation for downstream tasks and other numerical models. To investigate the effect of initial void ratio, two simulation systems were generated by performing different compaction conditions. Then, with a simplified DEM model, the authors firstly validate the physical properties, e.g. strain-stress relationships and shear modulus along the quasi-static loading, of generated particle systems using monotonic loading simulations. After that, groups of cyclic loading simulations under the constant volume condition were performed on the sand samples to reveal the mechanical responses of sand specimens with different void ratios. The degradation behaviors are represented as normalized degradation index versus number of loading cycles and characterized by analytical formulations.
The presented results provide some useful and interesting insights regarding the liquefaction of soil. In addition, as the author pointed out, the quantitative and analytical characterization of the degradation of strength and modulus could be helpful to other numerical simulations or even industrial applications. However, the manuscript needs quite some major revision before being published in Modeling. And I recommend that the manuscript has to address following points before reconsideration:
[a] A few minor points:
-
(P2, L47-48): ‘the numerical medium’ -> specimen?
-
(P4, L138-139): ‘The shear … by the following expression’ -> verb missing?
-
(P4, L141-142): ‘The incremental … rupture criterion’ -> I’m not sure about the logic here, could you give more clarification.
-
(P3-P4): Please use scientific notation in the equations and figures, especially to differentiate between scalar and vector.
-
(P4, L153): particle size is not a parameter of the model?
-
(P4, L156-158): I believe all these parameters have a direct and high impact on the properties simulated in your model. Could you rephrase what you mean here?
-
(P5, Eq.(5)): symbol r has not been mentioned or explained.
-
(P5, L161-162): why k needs to be higher than 1000 and why it needs to be negligible?
-
(P5, L162-163): why can mass be adjusted by Eq.15 and Eq.16. I think those two are the stability requirements of explicit integration.
-
(P7, L233-234): ‘a good similarity to real data’ -> a good agreement with experimental data?
-
(P7, L231): I’m a little puzzled by the deformation process, is it an isotropic compression followed by a simple shear? Give a configuration or visualization if possible.
-
(P7, Fig.3a): label of y-axis (deviatoric stress) is missing?
-
(P8, L259): why ‘loose sand shows a stronger nonlinearity’ is ‘as expected’?
-
(P10, L305): ‘in Figure6b is depicted the evolution’ -> Figure6(b) shows?
-
(P11, eq.17 and 18): symbols sigma_n^q and sigma_n^k have not been mentioned anywhere, I guess they are degradation indexes?
-
(P11, L338): ‘In Figure8b is depicted’
[b] Methodology & Discussion:
-
The author mentioned that the particle density in the model is a fictitious value of 10^12 kg/m^3 which is not physically realistic. I believe the particle mass is the key part of the governing equation and significantly affects the reactive force on particles and therefore stress response. Could the author give more detail about why this parameter value is feasible and please give more evidence of the mechanism if possible.
-
(P6, Fig. 2): This figure of a box of particles is not informative at all. Some information can be added, e.g. direction of coordinate, orthogonal? dimension of the system, which simulation system is this, loose or dense, loading direction, etc.
-
(P6, Table 1): What is the source of these parameters? Are they calibrated in this work or from some other literature? If they are calibrated, please give more details of the calibration. If from other literature, please give a citation.
-
(P6, L212-217): Could you give more details about how these two systems were created? E.g., dimension of the final systems, distribution of porosity, how are the relative densities calculated, etc. The characteristics of these two systems are the key of mechanical responses.
-
(P7, L238-250): I’m not sure how to connect the description in this paragraph with strain-stress curves in Fig.3. In Fig.3a, I’m seeing a lower deviatoric strength from the loose sample compared with the one in dense, so why do they ‘reach the same critical resistance’? How is the residual angle of friction and void ratio extracted from the figure? How is ‘large dilation’ being observed from the figure? Fig.3b is volumetric strain-stress, how is the ‘contractive and dilative behavior at various axial strain levels’ observed from Fig.3b?
-
(P9, L296-297): the author mentioned the degradation of loose sand is induced by ‘the increment of the excess pore pressure caused by the compaction of the soil grains’. Could you further explain how this is concluded from the simulation? Some extra information may be needed here, like the distribution/visualization of force or pressure on particles, how these values evolve over the deformation, etc.
-
(P13, Fig.10): The dashed lines and the equations, i.e., 0.002305*epsilon_zz^{-1.77} in both (a) and (b) are not explained. Figure 10(b) says it is for the medium-dense sample, but I don’t see there are epsilon_zz^{max} = 0.15%, 0.2% and 0.25% loading conditions that are shown in the figure legend.
-
The authors try to explain the macroscopic response, especially the stress response seen in the simulation, via the contractive and dilative behavior of grains and pore pressure change. However, no direct evidence is presented in the manuscript. I think more convincing information needs to be here to explain the mechanism of the strength and modulus evolution.
Round 2
Reviewer 1 Report
Again, The microscopic analysis and pictures are to prove the feasibility and accuracy of the simulation, otherwise,These curves will appear meaningless because they can be obtained through experiments.
What kind of sand is the object of this paper?, If not specified, use granular material?
Reviewer 2 Report
The quality of the paper is improved fundamentally by applying the comments and I think it is appropriate for publication. The only thing that I think is the English quality that should be enhanced in some parts.
Author Response
Thank you for your positive comment. The manuscript has been revised thoroughly and carefully. Errors and formatting issues have been corrected.
Reviewer 3 Report
The manuscript has been significantly improved. And the motivation, novelty and presentation of methodology and conclusion are sufficient to be accepted.
Author Response
Thank you very much for your positive comment and thoughtful review aimed at improving our manuscript.